# The (Patho)Biology of SRC Kinase in Platelets and Megakaryocytes

**DOI:** 10.3390/medicina56120633

**Published:** 2020-11-24

**Authors:** Lore De Kock, Kathleen Freson

**Affiliations:** Center for Molecular and Vascular Biology, Department of Cardiovascular Sciences, University of Leuven, 3000 Leuven, Belgium; lore.dekock@kuleuven.be

**Keywords:** SRC kinase, platelets, megakaryocytes

## Abstract

Proto-oncogene tyrosine-protein kinase SRC (SRC), as other members of the SRC family kinases (SFK), plays an important role in regulating signal transduction by different cell surface receptors after changes in the cellular environment. Here, we reviewed the role of SRC in platelets and megakaryocytes (MK). In platelets, inactive closed SRC is coupled to the β subunit of integrin α_IIb_β_3_ while upon fibrinogen binding during platelet activation, α_IIb_β_3_-mediated outside-in signaling is initiated by activation of SRC. Active open SRC now further stimulates many downstream effectors via tyrosine phosphorylation of enzymes, adaptors, and especially cytoskeletal components. Functional platelet studies using SRC knockout mice or broad spectrum SFK inhibitors pointed out that SRC mediates their spreading on fibrinogen. On the other hand, an activating pathological SRC missense variant E527K in humans that causes bleeding inhibits collagen-induced platelet activation while stimulating platelet spreading. The role of SRC in megakaryopoiesis is much less studied. SRC knockout mice have a normal platelet count though studies with SFK inhibitors point out that SRC could interfere with MK polyploidization and proplatelet formation but these inhibitors are not specific. Patients with the SRC E527K variant have thrombocytopenia due to hyperactive SRC that inhibits proplatelet formation after increased spreading of MK on fibrinogen and enhanced formation of podosomes. Studies in humans have contributed significantly to our understanding of SRC signaling in platelets and MK.

## 1. Introduction to SRC and SRC Family Kinases 

Over 100 years ago, the discovery of Rous sarcoma virus causing tumor growth in chickens, led to the characterization of the viral v-*SRC* gene [1]. This was followed by the discovery that v-SRC is a tyrosine kinase protein, essential for cell transformation by the Rous sarcoma virus, and derived from cellular chicken c-SRC, incorporated into Rous sarcoma virus via recombination [2]. Human proto-oncogene tyrosine-protein kinase SRC (SRC), the first proto-oncogene, was described as highly structurally similar to chicken c-SRC [3,4]. As v-SRC contains an altered carboxy terminus compared to SRC, these amino acids were expected to play an important role in the regulation of SRC activity and cellular transformation [5]. Human SRC contains two essential phosphorylation sites, namely tyrosine 419 (Y419, Y416 in chicken) and tyrosine 530 (Y530, Y527 in chicken) [6]. Intramolecular binding of phosphorylated Y530 to the SH2 SRC domain results in blocking of the catalytic site necessary for kinase activation and keeps SRC in a closed inactive state under normal cellular conditions [6,7] (Figure 1A). Y419 is located within the activation loop of the kinase domain and is conserved amongst many tyrosine kinases. After dephosphorylation of Y530 by protein tyrosine phosphatases [6], displacement of the SH2/SH3 domain opens up the kinase domain and allows (auto)phosphorylation of Y419, making SRC fully active [7] (Figure 1B). v-SRC lacks the carboxy-terminal domain including the Y527 self-inactivation residue and, therefore, v-SRC becomes constitutively active to stimulate malignant cell transformation [5]. Increased SRC activity is also detected in many human cancers, including several leukemias [8]. Pathogenic somatic *SRC* variants disturbing the SH2–Y530 interaction result in continuous SRC activation as found in colon and endometrial cancer [9,10]. 

SRC, similar to the other SRC family kinases (SFK), are nonreceptor kinases that are key regulators of signal transduction via diverse cell receptors to modulate cell differentiation, proliferation, survival, motility, and adhesion. SFK are structurally related and can be divided into two subfamilies: SRC-A family including SRC, YES, FYN, and FGR; SRC-B family including LYN, HCK, LCK, and BLK. Expression of SRC, YES, and FYN is present in almost all tissues, while the other SFK are mainly expressed in blood cells [11]. Although SRC expression is omnipresent throughout vertebrate cells, mainly osteoclasts, neurons and especially platelets present with the highest SRC protein expression compared to expression levels in other tissues [6,12]. This review will only focus on SRC kinase function in platelets and their bone marrow progenitor cells, megakaryocytes (MK), under normal and pathological conditions.

## 2. SRC Kinase Activity in Platelets in Health and Disease

### 2.1. SRC Kinase Regulates Platelet Activation

Platelets are activated to stop bleeding after vascular injury. This rapid activation response is mediated through various receptors that strongly rely on SFK activity. Of the eight family members, SRC, YES, FYN, FGR, LYN, HCK, and LCK are expressed in human platelets, while SRC, FYN, FGR, and LYN are expressed in mouse platelets [11]. SRC is the most abundant in human platelets, while in mouse platelets, LYN is the most abundant SFK member [11,13]. The high level of SRC kinase activity and expression in human platelets was already described in 1986 when the SRC protein was also found to be one of the major tyrosine phosphorylated proteins detected on platelet membranes [12]. 

Studies related to SRC function in platelets reveal a major role in integrin α_IIb_β_3_ signaling (Figure 2). The main platelet surface receptor integrin α_IIb_β_3_ (or glycoprotein IIbIIIa) initiates platelet ‘outside-in’ signaling that is essential for platelet aggregation and spreading. Integrin α_IIb_β_3_ remains inactive in resting platelets, while during ‘inside-out’ platelet activation, the α_IIb_β_3_ extracellular domain changes its conformation to an active and high-affinity form, thereby able to bind the ligands fibrinogen and von Willebrand’s factor (VWF) [14]. As the α and β cytoplasmic domains carry no intrinsic catalytic activity, the α_IIb_β_3_ receptor has to collaborate with diverse intracellular signaling proteins, including SRC that is coupled to the β_3_ subunit [11]. This interaction is accomplished by the SH3 domain of SRC and the C-terminal region of the β_3_ cytoplasmic tail and primes SRC for activation [15]. In resting platelets, C-terminal SRC kinase (CSK) keeps SRC in an inactive closed conformation by phosphorylating SRC-Y530 and it forms a complex together with SRC and the β_3_ subunit (Figure 2) [16]. Upon fibrinogen binding during platelet activation, β_3_ integrins cluster and α_IIb_β_3_-mediated outside-in signaling is initiated by activation of SRC through dephosphorylation of SRC-Y530 by protein tyrosine phosphatases, and dissociation of CSK followed by (auto)phosphorylation of SRC-Y418 in the activation loop [15,16,17]. The nonreceptor PTP-1B, recruited following fibrinogen binding to α_IIb_β_3_, is found to be essential for the dephosphorylation of SRC-Y530 and the subsequent initiation of outside-in signaling and platelet aggregation (Figure 2) [18]. 

Downstream effectors of α_IIb_β_3_-mediated outside-in signaling include many enzymes, adaptors, and cytoskeletal components [19]. α_IIb_β_3_-mediated outside-in signaling couples to actin polymerization and cytoskeletal reorganization to convert inactive discoid platelets to active fully spread platelets [14]. In the early stages of platelet spreading, prior to the appearance of lamellipodia and stress fibers, actin nodules are formed that participate in mechano-coupling between different platelets or platelets and other substrates. The generation of these F-actin rich structures is dependent on SRC activity and various actin cytoskeletal regulatory proteins stored in these actin nodules, such as the Wiskott–Aldrich syndrome protein WASP, the ARP2/3 complex, FYN, RAC1, talin, and vinculin [19,20]. Activation of SRC leads to the phosphorylation of tyrosine kinase SYK necessary to initiate cytoskeletal events for platelet spreading on fibrinogen (Figure 2) [16]. SYK can either interact with the α_IIb_β_3_ cytoplasmic tail or associate with the Fc receptor γ-chain homodimer (FcγRIIa) through its tyrosine-phosphorylated docking site within its immunoreceptor tyrosine-based activation motif (ITAM) [14,21]. In this way, further downstream phosphorylation of tyrosine kinase PYK2 connects the outside-in signaling pathway with class I phosphoinositide 3-kinase (PI3K), known to play a key role in thrombus stability via the small GTPase RAP1B (Figure 2) [14,22]. More SYK downstream substrates are the adaptor SLP-76 and VAV family proteins, all involved in cytoskeletal regulation [19]. Other proteins important for cytoskeletal regulation are phospholipase Cγ2 (PLCγ2) and focal adhesion kinase (FAK), which can directly be activated by SRC and other SFK and subsequently propagate signal transduction (Figure 2) [14]. PLCγ2 is involved in the formation of filopodia and lamellipodia during platelet spreading by mediating calcium signaling and PKC activity [23,24]. PLCγ2 can also indirectly be activated via SYK-PYK2 signaling [24]. The nonreceptor protein tyrosine kinase FAK can directly phosphorylate substrates including the actin-binding protein α-actinin [25], but can also act as a scaffold protein. Following integrin α_IIb_β_3_ activation, autophosphorylation of FAK on Tyr 397 provides a docking site for the SH2 domains of SFK, that in turn phosphorylate several FAK residues and then are able to recruit new binding partners such as GRB2, talin, paxillin, and vinculin (Figure 2). These proteins mediate the recruitment of FAK to focal adhesions, the sites where integrins interact with the extracellular matrix [26,27,28]. Talin has a dual role as this protein is important for integrin-cytoskeletal coupling, but can also directly regulate integrin affinity. The phosphorylation of Tyr 773 in the α_IIb_β_3_ cytoplasmic tail by SRC and other SFK negatively regulates talin binding to integrin α_IIb_β_3_ [29]. 

In addition to playing an important role in the activation of α_IIb_β_3_-mediated outside-in signaling and cytoskeletal regulation, SRC also participates in clathrin-mediated endocytosis of α_IIb_β_3_ during platelet activation via a SRC/PLC/PKC signaling pathway [30]. This is suggested because SRC, PLC, and PKC inhibitors block the membrane internalization of α_IIb_β_3_. Furthermore, SRC also participates in negative regulation of integrin α_IIb_β_3_ signaling by activating SH2 domain-containing inositol 5-phosphatase 1 (SHIP1) that associates with SRC, LYN, and α_IIb_β_3_ [31]. SHIP1 downstream signaling inhibits platelet spreading. Very recently, the adaptor molecule 14-3-3ζ was also found to directly interact with SRC and integrin α_IIb_β_3_ [32]. Together they form the 14-3-3ζ–SRC–β_3_ complex in platelets that regulates integrin signaling. During platelet activation, 14-3-3ζ increases SRC mobilization to integrin α_IIb_β_3_ and likely regulates its activation. 

SRC kinase is also involved in glycoprotein Ib-IX-V (GPIb-IX-V) signaling, necessary for binding of VWF near the damaged endothelium [33]. SRC can physically associate with GPIb and is activated by the interaction between GPIb and its ligand VWF, without being dependent on receptor clustering. Since the GPIb-IX-V receptor does not have tyrosine phosphorylated residues or other binding motifs, the association of GPIb and SRC is thought to be indirect via another signaling molecule [33]. PI3K was found to act as scaffolding protein that forms a complex together with GPIb and SRC in GPIb-stimulated platelets, and this complex appears to translocate to the cytoskeleton. Taken together, activation of SRC by GPIb is thought to be important for downstream tyrosine phosphorylation events [33].

Finally, SFK are also important for collagen receptor GPVI activation. GPVI forms a complex with the FcR γ-chain, whereby GPVI is able to bind calmodulin and the SH3 domains of SFK via a proline-rich domain in the GPVI cytoplasmic tail [34,35] (Figure 2). The receptor-like protein-tyrosine phosphatase (PTP) PTPRJ or CD148 was found to play an important role in regulating a pool of SRC at the plasma membrane of resting platelets and in regulating both collagen and FcRIIA-dependent signaling pathways [36]. SRC kinase-dependent tyrosine phosphorylation by primarily FYN and LYN takes place upon crosslinking of GPVI and induces activation of tyrosine kinase SYK and the downstream signaling cascade including adapters LAT, SLP-76, and PLCγ2. As FYN and LYN are palmitoylated they are able to reside in lipid rafts of the plasma membrane and therefore can mediate the phosphorylation of the FcR γ-chain. In contrast, SRC is not palmitoylated and is shown to be unable to induce this phosphorylation [34]. Therefore, SRC itself probably has no prominent role in GPVI-dependent platelet activation.

### 2.2. SRC Kinase Deficiency Impairs Platelet Function

Homozygous SRC knockout mice only survive the first weeks after birth and survivors present with impaired osteoclast function and osteopetrosis while they have no bleeding symptoms [37]. However, SRC^-/-^ platelets present with a markedly decreased tyrosine phosphorylation and spread poorly on fibrinogen [13,16]. Similar results were generated for FGR^-/-^, HCK^-/-^, and LYN^-/-^ mouse platelets. As there is a more pronounced inhibitory effect of spreading present when wild type platelets are treated with broad spectrum SFK inhibitors such as PP2, SU6656, and Dasatinib compared to what is observed for single SFK deficient mice, functional redundancy between SFK is expected as later also confirmed in double-deficient SFK mice [13]. However, SFK signaling during in vitro regulation of platelet-mediated monocyte recruitment in thrombo-inflammatory vascular disease is found to be nonredundant [38]. Mice lacking the integrin α_IIb_β_3_ residues necessary for the interaction between β_3_ and SRC also show defective platelet spreading on fibrinogen and reduced tyrosine phosphorylation of downstream SRC substrates [39]. CSK knockout mice show a large increase in platelet SFK activity including that of SRC [40]. Research using LYN^-/-^ SRC^-/-^ mice platelets show the presence of a greater delay in activation on collagen when compared to LYN^-/-^ mice platelets [13]. This suggests that SRC also plays a role in platelet activation by GPVI, but only in the absence of LYN, the major SFK that regulates GPVI signaling. In mice in vivo, Dasatinib increases the tail bleeding time in a dose-dependent and rapidly reversible manner [41].

Human platelets incubated with SFK inhibitors PP2 or SU6656 also fail to spread properly on fibrinogen and show defective phosphorylation of downstream substrates involved in cytoskeletal regulation [16]. Ex vivo, in whole blood of patients treated with Dasatinib under physiologic flow, less and smaller thrombi are formed compared to healthy controls [41]. Inactivating pathogenic *SRC* variants in humans have not yet been described. On the contrary, the germline heterozygous gain-of-function (GOF) SRC variant E527K is detected in patients from three unrelated families who present with thrombocytopenia and a paucity of alpha granules, clinical bleeding symptoms, and more variable phenotypes that include myelofibrosis, osteoporosis, facial dysmorphism, and behavior defects [42,43,44]. The substitution of glutamic acid (E) by lysine (K) at residue 527 lies in close proximity to the inhibitory Y530 phosphorylation site and therefore is shown to interfere with the self-inhibitory capacity of SRC, causing constitutively active SRC activity in platelets (Figure 1B) [42]. Given the prominent role of SRC in α_IIb_β_3_-mediated outside-in signaling while this SFK would not influence GPVI signaling, it is remarkable that E527K deficient platelets present with a reduced collagen-induced aggregation while normal responses were noticed for agonists as ADP, TRAP, U46619, and arachidonic acid [42,43,44]. Further platelet function studies are required to pinpoint this collagen signaling defect caused by an GOF SRC variant. E527K deficient platelets also show increased adhesion and spreading on fibrinogen, type I collagen, and VWF, and they have increased active FAK protein expression compared to healthy controls [44].

High SFK activity is also detected in diverse human cancers. Therefore, tyrosine kinase inhibitors (TKI) are often used as a targeted therapy for cancer treatment [45]. To date, 38 TKI inhibitors have already been developed and though most have a relatively high affinity for the targeted SFK, they are never fully specific. As described above, platelet activation is strongly dependent on SFK activity and therefore platelet-related side effects during cancer treatment with TKI, such as thrombocytopenia and the inhibition of platelet function, are often present, resulting in an increased bleeding risk [45]. 

## 3. SRC Kinase activity in Megakaryocytes in Health and Disease

### 3.1. SRC Kinase Regulates Megakaryopoiesis

Megakaryopoiesis is the differentiation from hematopoietic stem cells to MK in the bone marrow. During maturation, megakaryoblasts undergo endomitosis, a process by which DNA replicates without cell division, and therefore become polyploid and alpha and dense granules are formed [46,47]. MK have to migrate from the osteoblastic niche towards the sinusoidal wall of the blood vessel via interactions with extracellular matrix proteins of the osteoblastic and vascular niche. Podosomes play an important role as they degrade matrix proteins and can protrude through the basement membrane [48]. When actin reorganization takes place, late MK will start to form proplatelets via long cytoplasmic extensions and will eventually release platelets into the blood circulation. Many cytoskeletal components such as microtubules and actin filaments are important regulators as organelles and platelet granules have to be transported along the proplatelet extensions [46]. Defects in megakaryopoiesis can result in decreased platelet production or thrombocytopenia.

MK express high levels of SRC, FYN, LYN, and SYK [46]. Although SRC is intensively studied in platelet activation, its specific role during megakaryopoiesis and platelet formation still remains largely unknown. There is some experimental evidence that SRC plays a role in MK polyploidization and granule formation during maturation, as well as in proplatelet formation (Figure 3) [49,50,51,52,53] but such evidence comes from SFK inhibition studies and these provide a rather indirect evidence (see further). Cortactin, a filamentous actin-binding protein and substrate of SRC, is expressed in bone marrow MK (Figure 3). The interaction between cortactin and SRC is thought to play a role in MK maturation as upregulation of SRC during MK maturation in murine tissues is followed by upregulation of cortactin expression [54].

### 3.2. SRC Kinase Deficiency Impairs Megakaryopoiesis 

Mature MK express high levels of integrin α_IIb_β_3_ and this is essential for proplatelet formation on fibrinogen, which is present in the vascular niche [55]. SFK inhibition of murine bone marrow aspirates with the broad spectrum SFK inhibitor PP2 elevates proplatelet formation and this suggests that α_IIb_β_3_-mediated outside-in signals are not required as they rely on SFK activity [55]. A novel α_IIb_β_3_-mediated SRC-SYK-PLCgamma2 signaling pathway would play an important role in the regulation of spreading and migration of MK on fibronectin and in platelet formation [56]. Treatment with the SFK inhibitor PP1 abolishes phosphorylation of PLCgamma2 in mouse primary BM-derived MK. As cell–cell and cell–matrix interactions have an impact on MK differentiation, also FAK is thought to be involved in megakaryopoiesis [27]. FAK, a previously discussed downstream substrate of SRC involved in cytoskeletal regulation, is discovered to negatively regulate megakaryopoiesis through inhibition of thrombopoietin (TPO) signaling, since MK lineage-specific FAK knockout mice show more MK progenitors and mature MK with higher polyploidization [27]. 

In vivo administration of Dasatinib, a broad spectrum TKI, in mice causes a 30% reduction in platelet count, which is similar to the mild thrombocytopenia present in chronic myeloid leukemia (CML) patients treated with Dasatinib [53]. These data suggest that Dasatinib inhibits platelet formation. On the contrary, the MK count in the bone marrow is elevated together with the polyploidization. Mature MK collected from mouse bone marrow, further cultured and incubated with Dasatinib and plated on a fibronectin-coated surface, show increased differentiation but reduced migration and proplatelet formation [53]. In contrast, another study shows that platelet production in mice infused with the SFK inhibitor SU6656 is increased [51]. Enhanced polyploidization and maturation, indicated by ploidy, size, and granularity of MK, is present when culturing human MK cell lines, bone marrow progenitors, and CD34+ stem cells together with SFK inhibitors [49,50,51,52]. As all TKI target multiple SFK and some SFK have contradictory effects, specific SRC inhibitors are required to define its role in megakaryopoiesis. SRC knockout mice have a normal platelet count [37].

The only direct evidence for an important role of SRC in megakaryopoiesis originates from the observation that patients with the GOF E527K variant present with thrombocytopenia [42,43,44]. Continuously active SRC in MK from the patients is shown to inhibit proplatelet formation that could partially be restored by adding an SFK inhibitor to these cultures [42]. In addition, E527K deficient MK present with increased spreading on fibrinogen with a reduction in proplatelet formation [44]. E527K deficient MK also present with an increased number and density of podosomes [42,44]. This suggests that the thrombocytopenia seen in SRC E527K patients could be (partially) due to an altered interaction of MK with the extracellular matrix [44]. The E527K patients also present with a lack of α-granules and this suggests that overactive SRC alters downstream pathways involved in α-granule formation, however, the underlying mechanism for this defect in granulopoiesis remains unknown. Studies of downstream effectors are required to understand all these observations.

## 4. Conclusions

The role of SRC in platelet biology has been extensively studied and reveals its important role in integrin α_IIb_β_3_ outside-in signaling upon fibrinogen binding. Upon activation, SRC stimulates many downstream substrates, especially cytoskeletal components necessary for platelet spreading and aggregation. SRC knockout mice show no reduced platelet count or bleeding symptoms but their platelets spread poorly on fibrinogen [13,16]. On the contrary, proplatelet formation is impaired when SRC is hyperactive in MK due to the genetic variant E527K in three unrelated pedigrees with inherited thrombocytopenia [42,43,44]. SRC hyperactivity in platelets from these patients was shown to result in higher tyrosine phosphorylation of multiple platelet proteins, including FAK, which plays an important role in cytoskeletal reorganization but also in the interaction of MK with extracellular matrix proteins [42,43,44]. However, the role of SRC in megakaryopoiesis requires further studies to define the underlying pathways. SRC E527K deficient MK show increased spreading on fibrinogen and podosome formation. The interaction between MK and the bone marrow matrix components is thought to be essential for proplatelet formation and defective SRC regulation therefore probably lies at the basis of impaired proplatelet formation due to defective interaction with matrix proteins resulting in thrombocytopenia [44]. Studies in humans have contributed significantly to our understanding of SRC signaling in platelets and MK.

## Figures and Tables

**Figure 1 medicina-56-00633-f001:**
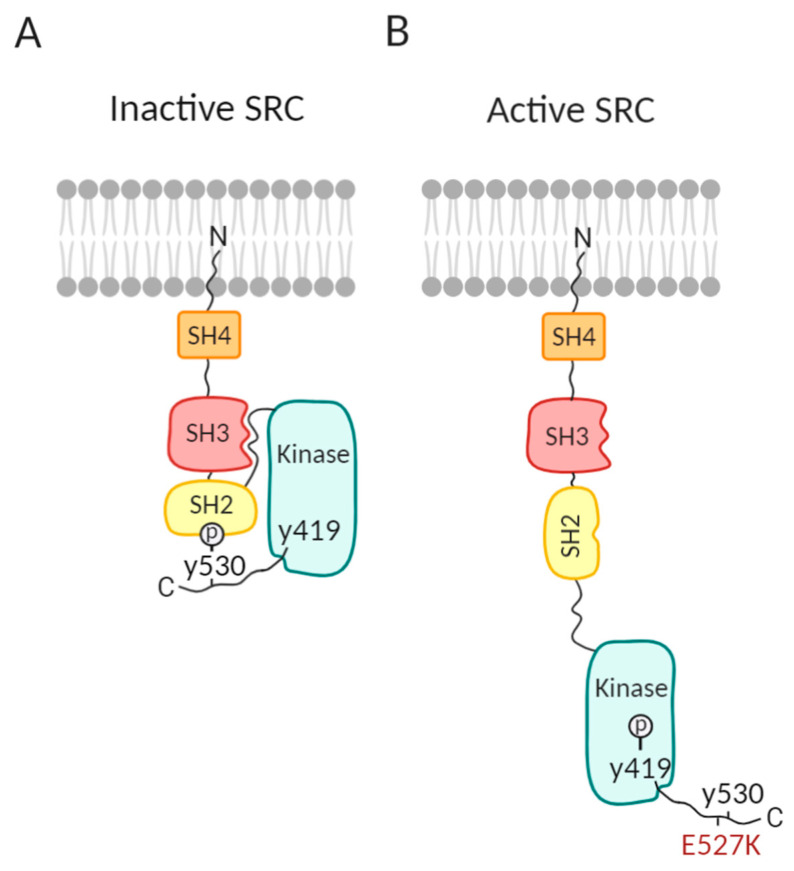
Proto-oncogene tyrosine-protein kinase SRC (SRC) changes its conformation upon activation. (**A**) Closed inactive SRC under normal cellular conditions. Y530, located in the C-terminal tail of SRC, is phosphorylated and binds to the SH2 SRC domain, thereby blocking the catalytic site necessary for kinase activation. (**B**) Active SRC. When Y530 is dephosphorylated, displacement of the SH2/SH3 domains allows (auto)phosphorylation of Y419 and activation of SRC. Variants located near Y530, such as the germline E527K variant, disturb the SH2–Y530 interaction and also result in continuous SRC activation.

**Figure 2 medicina-56-00633-f002:**
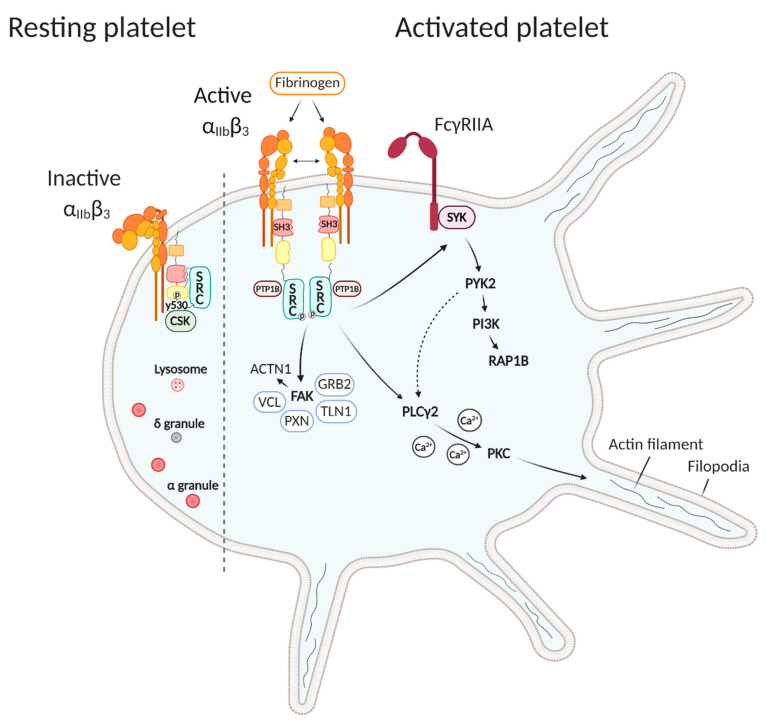
The role of SRC kinase during platelet activation. Left side: In circulating resting platelets, SRC is maintained in an inactive closed conformation, mediated through Y530 phosphorylation by C-terminal SRC kinase (CSK), and is coupled to the β subunit of integrin α_IIb_β_3_. Right side: Fibrinogen binding initiates integrin α_IIb_β_3_ dimerization and platelet activation. After dephosphorylation of Y530 by protein-tyrosine phosphatases (PTP) and (auto)phosphorylation of Y419, SRC becomes active and initiates downstream outside-in signaling. SRC activates the focal adhesion kinase FAK, that in turn activates α-actinin (ACTN1) or recruits new binding partners such as GRB2, talin (TLN1), paxillin (PXN), and vinculin (VCL), which mobilize FAK to focal adhesions. SRC can also phosphorylate PLCγ2, which plays an important role in the formation of filopodia and lamellipodia during platelet spreading by mediating calcium signaling and PKC activity. Additionally, SRC can phosphorylate tyrosine kinase SYK, coupled to the FcγRIIa receptor, that in turn phosphorylates tyrosine kinase PYK2, which further activates PLCγ2 or connects the outside-in signaling pathway with class I phosphoinositide 3-kinase (PI3K) to mediate thrombus stability via the small GTPase RAP1B.

**Figure 3 medicina-56-00633-f003:**
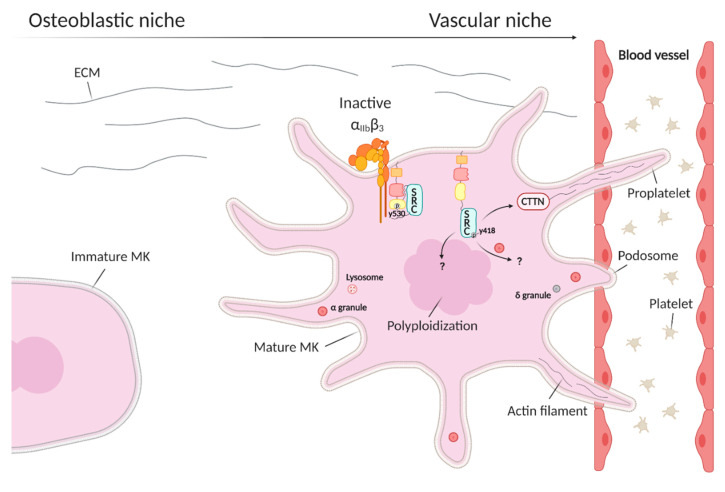
The role of SRC kinase during megakaryopoiesis. During megakaryocyte (MK) maturation, MK migrate from the osteoblastic to the vascular niche by interacting with extracellular matrix (ECM) proteins. The mature MK use podosomes to migrate from the osteoblastic to the vascular niche and through the vascular wall, to extend proplatelets and finally release platelets into the blood circulation. Opposite to its role in platelets, the integrin α_IIb_β_3_ outside-in signaling pathway is not active in maturating MK. SRC can activate the actin-binding protein cortactin (CTTN) during MK maturation.

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
