# Peer review of "The (Patho)Biology of SRC Kinase in Platelets and Megakaryocytes"

_medicina, 2020, doi:10.3390/medicina56120633_

Round 1

Reviewer 1 Report

De Kock and Freson present a concise and well-written short review on the SRC kinase biology in platelets and megakaryocytes.  It is balanced, well-referenced and up-to-date, and enjoyable to read.

Minor suggestions are found below:

  1. Consider adding information on non-redundancy of SFK isoforms in thrombo-inflammation (e.g. PMID 29974666).
  1. Does the fact that E527K patients present with a lack of α-granules support the role of SRC in granule formation in MKs? Please discuss briefly.
  1. Consider adding a section on the involvement of SRC kinase in the glycoprotein complex Ib-IX-V signaling (e.g. PMID 12393736 and others).
  2. Please harmonize abbreviations from 1st mention (e.g. MK, PTP, etc).
  3. Title of section 3, please correct “impaires” with “impairs”

Reviewer 2 Report

This review article from L. De Kock and K. Freson focuses on the (patho)biology of Src kinases in platelets/megakaryocytes. This group has identified in 2019 a family of patients exhibiting a pathological gain of function Src mutation leading to thrombocytopenia and bleeding.

This is a very good review which summarizes the recent findings on the role of Src kinases in platelet and megakaryocytes and the impact of a gain of function mutation identified in patients. I have only a few remarks:

 Specific points:

  • Figure 2 and text: GPVI is associated with the FcR gamma-chain not with FcgRIIa, please clarify this point.
  • Some references showing the impact of Src inhibition on platelet in vivo (Blood 2009, 114, 1884-92), the potential role of CD148 on Src kinase activation (Blood 2012, 120, 1309-16) and the interaction of the SH3 domains of Src kinases with the proline-rich domain of GPVI (J Biol Chem 2002, 277, 21561-66) should be included.

Minor points:

  • Some typos to be corrected: “ 3. SRC kinase deficiency impairs platelet function”, etc…
